# Toward a Personalized Therapy in Soft-Tissue Sarcomas: State of the Art and Future Directions

**DOI:** 10.3390/cancers13102359

**Published:** 2021-05-13

**Authors:** Liliana Montella, Lucia Altucci, Federica Sarno, Carlo Buonerba, Stefano De Simone, Bianca Arianna Facchini, Elisena Franzese, Ferdinando De Vita, Salvatore Tafuto, Massimiliano Berretta, Gaetano Facchini

**Affiliations:** 1Naples 2 Local Health Unit, Oncology Operative Unit, “Santa Maria delle Grazie” Hospital, 80078 Naples, Italy; stefano.desimone@aslnapoli2nord.it (S.D.S.); elisena.franzese@aslnapoli2nord.it (E.F.); gaetano.facchini@aslnapoli2nord.it (G.F.); 2Precision Medicine Department, “Luigi Vanvitelli” University of Campania, 80138 Naples, Italy; lucia.altucci@unicampania.it (L.A.); federica.sarno@unicampania.it (F.S.); 3Regional Reference Center for Rare Tumors, Department of Oncology and Hematology, AOU Federico II of Naples, 80131 Naples, Italy; carlo.buonerba@izsmportici.it; 4National Reference Center for Environmental Health, Zoo-Prophylactic Institute of Southern Italy, 80055 Portici, Italy; 5Division of Medical Oncology, Precision Medicine Department, “Luigi Vanvitelli” University of Campania, 80131 Naples, Italy; biancaarianna.facchini@studenti.unicampania.it (B.A.F.); ferdinando.devita@unicampania.it (F.D.V.); 6Sarcomas and Rare Tumours Unit, Istituto Nazionale Tumori, IRCCS Fondazione G. Pascale, 80131 Naples, Italy; s.tafuto@istitutotumori.na.it; 7Department of Clinical and Experimental Medicine, University of Messina, 98121 Messina, Italy; massimiliano.berretta@unime.it

**Keywords:** sarcoma, precision medicine, personalized medicine, translocation, genome

## Abstract

**Simple Summary:**

Soft-tissue sarcomas encompass heterogeneous histotypes with variable clinical behavior. The cornerstone of treatment is represented by surgery when the disease is diagnosed at an early stage. However, in recurrent and metastatic stages, conventional available therapeutic options yield disappointing results. In the era of precision medicine characterized by exciting advancements in several malignancies, soft-tissue sarcoma treatment still represents an unmet need.

**Abstract:**

Soft-tissue sarcomas are rare tumors characterized by pathogenetic, morphological, and clinical intrinsic variability. Median survival of patients with advanced tumors are usually chemo- and radio-resistant, and standard treatments yield low response rates and poor survival results. The identification of defined genomic alterations in sarcoma could represent the premise for targeted treatments. Summarizing, soft-tissue sarcomas can be differentiated into histotypes with reciprocal chromosomal translocations, with defined oncogenic mutations and complex karyotypes. If the latter are improbably approached with targeted treatments, many suggest that innovative therapies interfering with the identified fusion oncoproteins and altered pathways could be potentially resolutive. In most cases, the characteristic genetic signature is discouragingly defined as “undruggable”, which poses a challenge for the development of novel pharmacological approaches. In this review, a summary of genomic alterations recognized in most common soft-tissue sarcoma is reported together with current and future therapeutic opportunities.

## 1. Introduction


Soft-tissue sarcomas (STS) are rare tumors representing around 1% of all adult malignancies. They include more than 50 heterogeneous subtypes of tumors deriving from mesenchymal cells. Despite the diverse behavior shown by the different subtypes, most tumors are aggressive and have a high rate of local recurrence and distant metastases. Median survival of patients with advanced STS reaches approximately 20 months, with poor response to standard treatments [1].

Radical surgery is the mainstay of treatment in localized disease. Locally advanced or metastatic disease is usually treated with chemotherapy and/or radiotherapy, although the prognosis is poor because of primary or secondary chemo- and radio-resistance. Radiotherapy plays a definite role in several settings of STS. In adjuvant settings, radiotherapy may reduce the recurrence risk especially when there are close or infiltrated margins. In the neoadjuvant setting, the combined use of radio- and chemo-therapy produced better results in terms of overall survival in high-risk STS of the extremities [2]. Stereotactic body radiotherapy compares well to surgery in case of lung metastases [3]. In an advanced/palliative setting, radiotherapy may represent a compelling choice.

The backbone of chemotherapy is represented by regimens including anthracyclines. Single-agent doxorubicin is associated with a 10–30% overall response rate and a median overall survival of 8–17 months [4]. Anthracycline-based combination regimens have yielded increased response rates and progression-free survival (PFS) without overall survival (OS) improvements and at the expense of increased toxicity [5]. Other active drugs used in STS are ifosfamide, dacarbazine, gemcitabine, and docetaxel.

Trabectedin and pazopanib represent two significant advancements in the drug arsenal of sarcomas. Trabectedin (ET743, Yondelis^®^) derives from the marine ascidian, Ecteinascidia turbinate, and is now manufactured synthetically. It binds to the minor groove of DNA, disrupting cell-cycle progression and inhibiting cell proliferation. Approval was based on the results of a pivotal Phase III trial with a 2:1 randomization of 518 patients with advanced liposarcoma (LPS) and leiomyosarcoma (LMS) in which a significant improvement in PFS was reported in the trabectedin group vs. the dacarbazine group after the failure of prior chemotherapy [6]. Initial information coming from a compassionate-use program of trabectedin documented activity in advanced pretreated myxoid LPS (MLPS) [7]. Since trabectedin approval in Europe in 2007, an expanding amount of data has been supporting its efficacy in real-world settings [8]. Recent data confirm the efficacy of trabectedin in patients with LPS and LMS with higher PFS in MLPS [9].

In 2012, results of the Palette study opened the avenue to targeted therapies in STS. Pazopanib was compared to placebo in metastatic STS, progressing after previous standard chemotherapy. Pazopanib is a multitargeted tyrosine kinase inhibitor (TKI) with preferential activity on vascular endothelial growth factor (VEGF) receptor 1,2,3, platelet-derived growth factor (PDGF) receptor α and β, c-Kit, and, at a minor level, fibroblast growth factor (FGF) receptor 1 and 3. A significant improvement in PFS was documented in pazopanib-treated patients [10].

On January 28, 2016, the FDA approved eribulin (Halaven; Eisai Inc., Tokyo, Japan) for the treatment of patients with unresectable or metastatic LPS who have received a prior anthracycline-containing regimen. This approval was mainly based on a two-month advantage in OS shown by a randomized study comparing eribulin to dacarbazine in advanced or metastatic pretreated STS [11]. It must be noted that in all reported studies, dacarbazine was chosen as a comparative arm based on an overall response rate (ORR) as a single agent of 16–20% and a median PFS of 2 months [12,13].

In summary, one of the cited studies showed a limited benefit in OS, while the others presented significant improvements in median PFS only, which highlights the compelling and unmet need for active and effective novel treatments in STS.

The sunrise of precision medicine in 2000 fed hopes that advancements reported in tumors, such as lung cancer, could be accomplished in other malignancies, as well. Olaratumab, a monoclonal antibody against the PDGFRα, may represent a new, rationale-based, perspective in STS, given its antiangiogenic properties useful in tumors with marked angiogenesis. It received accelerated approval by the FDA in 2016 based on preliminary results of a phase 1b, randomized, phase 2 study [14]. However, the phase III trial did not confirm any advantage over doxorubicin alone [15], which smothered the initial clamors.

In other cases, such as gastrointestinal stromal tumors (GIST), the identification of driver mutations in KIT and PDGFRα have effectively realized tailored therapies which radically changed tumor history.

From a simplistic perspective, STS might be classified into three subclasses: with reciprocal chromosomal translocations, with defined oncogenic mutations and with complex karyotypes. Detection of translocations and fusion proteins suggests a role for interferences with the related pathways. However, in most instances, this logical mindset does not reflect in the identification of molecules showing promising results in preliminary studies. Most drugs selected through intensive preclinical screening do not successfully pass through phases I and II due to toxicity or poor activity, and only a few phase III studies are available for STS, also because of difficulties in accrual.

In many cases, a wide range of genomic aberrations was found, and no druggable mutations were identified. STS can be compared to a matryoshka: as the number of studies on some genetic events increases, other nondriver, stochastic aberrations are found, together with mutations in coding regions, epigenetic changes, and secondary mutations.

Taking into account these premises, we review here the actual knowledge concerning the basic background of most common adult STS with a view on clinical impact. After that, we summarize clinical ongoing studies and paint possible future landscapes, realizing that the present remains a partial outlook of a complex framework.

## 2. Liposarcoma (LPS)


LPS is the most common STS, representing 15–25% of all STS [16,17].

LPS is derived from adipocytic cells and is classified according to 2020 WHO Classification [18] into four types: well-differentiated (WDLPS), dedifferentiated (DDLPS), myxoid (MLPS), and pleomorphic LPS (PLPS) (Table 1). They usually arise in the limbs or retroperitoneum, with DDLPS being more frequent in the retroperitoneum and PLPS in the limbs. The most frequent types of LPS are WDLPS and DDLPS. From a morphological point of view, WDLPS resembles the normal counterpart, DDLPS is characterized by high-grade representative cells, and MLPS has a predominantly myxoid stroma.

### 2.1. Genomic Alterations in LPS


LPS has extensive studies of the genomic alterations and represents a good example of the heterogeneous and complex aberrations found in the different subtypes.

### 2.2. WDLPS and DDLPS


Chromosome 12 is the hallmark of WDLPS and DDLPS (
Figure 1
A).

WDLPS and DDLPS show 12q13-15 amplification (http://atlasgeneticsoncology.org/Genes/GC_CPM.html, accessed on 28 February 2021) (Figure 2, Table 2). This feature appears cytogenetically with supernumerary ring or giant rod chromosomes [16,18]. The amplified DNA stretch corresponds to various cancer-related genes, the most studied being MDM2 (or HDM2 in humans) and CDK4, but also HMG2A, TSPAN31, YEATS4, and CPM [16,17,19].

Mdm2 is an oncoprotein that blocks p53 tumor-suppressor-mediated transcriptional transactivation, guides p53 from the cell nucleus to the cytoplasm, and polyubiquitylates p53 (Figure 1B). Mdm2 is an important negative regulator of the p53 tumor suppressor and is amplified in approximately 100% of WDLPS and DDLPS. The presence of mdm2 is roughly equivalent to functional p53 inactivation. The amplification of MDM2 balances the lack of p53 mutations, which are found only in 10–20% as compared to 60% of PLPS. MDM2 presence and p53 mutation translate into proliferation and tumor aggressiveness [16]. Selective pressure induced by MDM antagonists induces emerging p53 mutations that determine resistance to treatment [20].

CDK4 is a key regulator of the G1/S cell-cycle checkpoint. Together MDM2 and CDK4 are amplified in over 90% of patients.

HMG2A, also known as HMGA2, is a protein that belongs to the nonhistone chromosomal high-mobility group (HMG) protein family that is involved in the regulation of transcription, replication, recombination, and DNA repair [21]. HMG2A has a diagnostic and prognostic value. In fact, amplification of HMGA2 was associated with the atypical lipomatous tumor/well-differentiated liposarcoma histological type and a good prognosis, whereas CDK4 and JUN amplifications were associated with DDLPS histology and a bad prognosis [22]. Indeed, MDM2/HMGA2 amplification or gain ratio was found to have a significant prognostic value [23].

TSPAN31 is a member of the transmembrane 4 superfamily, also known as the tetraspanin family. Most of these members are cell-surface proteins characterized by the presence of four hydrophobic domains. These proteins mediate signal transduction events that play a role in the regulation of cell development, activation, growth, and motility.

YEATS4, also known as GAS41, is a nuclear protein encoded by the GAS41 (glioma-amplified sequence) [24] gene that was identified in a glioblastoma cell line. It is a member of the YEATS family of proteins and is implicated in chromatin remodeling and transcriptional regulation [25].

CPM is an enzyme able to perform cleavage of C-terminal arginine or lysine residues from polypeptides, thus inducing activation of growth factors [26]. Interestingly, the *CPM* gene is located downstream from MDM2. CPM amplification differentiates WDLPS from benign variants [26].

FGF receptor (FGFR) substrate 2 (FRS2) is located on chromosome 12q13-15, which is frequently amplified in liposarcomas as previously outlined [27]. FGRFR pathway seems to be relevant in DDLPS, and this role could have therapeutic implications.

### 2.3. Aurora Kinases in LPS


Aurora kinases are serine/threonine kinases essential for cell proliferation and distribution of genetic materials to its daughter cells. AURKA is significantly upregulated in DDLPS, compared with WDLPS [28]. These premises suggest a role for Aurora kinases as a therapeutic target.

### 2.4. MLPS


In MLPS, other genetic features involving chromosome 12 are found (Figure 2, Table 2). In about 95% of cases the translocation t(12;16)(q13;p11) (Figure 1C) producing FUS (fused in sarcoma) -DDIT3 (DNA damage-inducible transcript) (otherwise FUS-CHOP) fusion protein is found [19]. In the remaining 5% of cases, the translocation t(12;22)(q13;q12) producing EWSR1 (Ewing sarcoma breakpoint region 1)-DDIT3 fusion protein is recognized [19].

FUS and EWSR1 belong to the FET group of fusion oncogenes, found primarily in sarcomas and leukemias. The FET gene fragments are juxtaposed to one of several alternative transcription factor (TF)-encoding genes. The fusion gene products invariably consist of an N-terminal domain (NTD) derived from one of the FET proteins fused with the DNA binding domain from the TF partner. FET family members can bind RNA displaying regulatory functions and are involved in genomic integrity.

The location of the FUS gene has been identified at 16p11 by the site of the breakpoint in the translocation. The FUS protein contains an RNA-recognition motif and is a component of nuclear riboprotein complexes. Lack of FUS in mice causes lethality within the neonatal period, influences lymphocyte development and proliferative responses of B cells to specific mitogenic stimuli. FUS is important in genome maintenance. The DDIT3 gene is located on chromosome 12 (12q13.1-q13.2). The related protein CHOP (C/EBP-homologous protein) is a nuclear protein, with a key role in stress response.

The FUS-DDIT3 fusion protein is considered an abnormal TF.

Transformation of primary cells by FUS-DDIT3 has been demonstrated in mouse mesenchymal progenitor cells (MPCs) that form MLPS-like tumors when introduced in SCID mice [29]. The causal role of FUS-DDIT3 in MLS development has been further demonstrated in transgenic mice [30].

FUS-DDIT3 induced aberrant IGF-IR/PI3K/Akt pathway activity, dependent on transcriptional induction of the IGF2 gene [31].

In a case series, 26.8% of MLPS cases displayed activating alterations in PI3K/Akt signaling components, predominantly PIK3CA gain-of-function mutations [32].

Mesenchymal progenitor cells carrying FUS-CHOP fusion protein show induction of growth factors such as PDGFA, HGF, cytokines (IL-6), MET receptor, and cell-cycle regulators (CDK4, MDM2) [16].

### 2.5. PLPS


PLPS is the rarest LPS. It has complex karyotypes and a high frequency of p53 mutations (60%).

### 2.6. microRNAs in LPS


A booming field in sarcoma is represented by microRNAs (miRNAs or miRs), noncoding RNAs regulating target gene expression and thus influencing many cell functions, including proliferation, apoptosis, differentiation, oncogenesis, and drug resistance in malignant cells. The dysregulation of miRs is involved in the initiation and progression of human cancers, including LPS [33]. As an example, miR-193b was significantly underexpressed in DDPLS compared to normal adipose tissues. miRNA may define different liposarcoma subtypes [34]. Reintroduction of miR-193b induced apoptosis in liposarcoma cells and promoted adipogenesis in human adipose-derived stem cells [35]. miR-143, miR-145, and miR-451 act as tumor suppressors in adipose tissue, and re-expression of these miRNAs may represent a promising therapeutic strategy for liposarcomas [34,36].

### 2.7. Notch Pathway in LPS


Notch is included among the pathways that are considered crucial for the regulation of cell fate [37]. Canonical Notch signaling controls both embryonic and adult stem cell self-renewal, stem cell quiescence, cell fate and differentiation, cell survival, apoptosis, and tumorigenesis. The Notch receptor is classified as a large single-pass type 1 transmembrane glycoprotein. It is expressed as a heterodimer at the cell membrane. Preclinical studies showed a role for the Notch pathway in LPS pathogenesis [37]. There are different strategies in targeting the Notch signaling pathway, including the development of monoclonal antibodies against the Notch transmembrane receptors or gamma-secretase complex proteolytic activity (c-secretase) inhibitors that prevent the release of the Notch intracellular domain and thus its translocation to the nucleus where it mediates its main activity [38].

### 2.8. Innovative Therapeutic Approaches in LPS


Table 3 reports the most relevant clinical completed and ongoing studies.

The previously described amplification of genes located in chromosome 12 as well as the pathognomonic translocation found in MLPS, which both appear determinant in LPS pathogenesis, has not yet translated into significant clinical progress.

Starting with a rationale-based interpretation, MDM2 antagonists may represent active drugs in LPS. Early-phase clinical trials of MDM2 antagonists showed evidence of antitumor activity in patients with leukemia and liposarcoma [40].

The pan-FGFR inhibitor erdafitinib reduced cell viability and induced apoptosis by strongly inhibiting the ERK1/2 pathway. The combination of erdafitinib with the MDM2 antagonist RG7388 exerted a synergistic effect [41].

Nutlins inhibit the interaction between MDM2 and tumor suppressor p53 [42]. Nutlins fill the p53 binding pocket of MDM2 and effectively disrupt the p53–MDM2 interaction that leads to activation of the p53 pathway in p53 wild-type cells. Nutlin-3 is the compound most used in anticancer studies. RG7112, a derivative of nutlin-3, was preliminarily tested in a study enrolling patients with unresected lesions to derive pathological and clinical information. In biopsies from treated patients, p53 and p21 expression increased by three-fold. One partial response and stable disease in 14 out of 20 patients have been reported, with neutropenia and thrombocytopenia being the most reported adverse events [43].

In a phase I study testing SAR405838, another MDM2 inhibitor, the best response was stable disease in 56%, and the progression-free rate at 3 months was 32% [44]. Patient selection (p53 wild-type) and p53 mutation as a mechanism of resistance appear to be critical and presumably influence low response rates.

MK-8242 is another small MDM2 inhibitor that has shown partial response and prolonged progression-free survival in a phase I study [45]. However, to the best of our knowledge, no further investigation of this drug is ongoing [39].

Attempts to overcome resistance are also addressed by combination strategies, for example, the combination of MDM2 antagonists and MEK inhibitors [46].

Given the previously described role for CDK4, another class of drugs potentially useful in LPS is represented by CDK4 inhibitors. In patients with advanced WDLPS/DDDLS, treatment with palbociclib was associated with a favorable PFS and occasional tumor response [47]. The potential role for palbociclib is now being investigated in the second-line setting in the PalboSarc phase II trial (NCT03242382), which enrolls advanced sarcomas with overexpression of CDK4 but excludes LPS.

Single-agent ribociclib does not seem to achieve significant results [48]. A combination of ribociclib and everolimus has been tested in a phase II ongoing study (NCT03114527) enrolling advanced DDPLS and LMS.

Combo strategies hold the promise to be more successful and have been tested. CDK4 inhibitors can act as potentiators of MDM2 antagonists in DDLPS at least in xenograft models [49]. A phase II study of CDK4/6 inhibition with palbociclib combined with programmed cell death protein 1 (PD-1) blockade (INCMGA00012) (NCT04438824) in patients with advanced WDLPS and/or DDLPS is underway.

Selinexor, a selective inhibitor of nuclear export (SINE), seems to arrest the cell cycle and leads to apoptosis. In December 2020, it was approved by the FDA in combination with bortezomib and dexamethasone for the treatment of adult patients with pretreated multiple myeloma. A phase 2–3, multicenter, randomized, double-blind study of selinexor (KPT-330) vs. placebo in patients with advanced unresectable DDLPS (NCT02606461) has been performed, confirming the disease stabilization shown in the preliminary investigations [50]. Recruiting trials with selinexor are now investigating combination with gemcitabine in selected advanced STS and osteosarcoma (NCT04595994) and with imatinib in GIST (NCT04138381).

Among the ongoing studies enrolling LPS, there is also a phase II study with sitravatinib (MGCD516), which is a small-molecule inhibitor of multiple tyrosine kinases.

## 3. Leiomyosarcoma (LMS)


LMS derives from smooth-muscle connective tissues and represents 5–10% of soft-tissue sarcomas [51]. Given the wide presence of smooth muscle throughout the body, LMS can arise anywhere, but most commonly, it occurs in the uterus (ULMS). Unlike other sarcomas, LMS does not show a single key alteration that is determinant in pathogenesis; hence, it has been also defined as “nontranslocation-related sarcoma”, together with undifferentiated pleomorphic sarcoma and myxofibrosarcoma. p53 and RB1 are the two main pathways altered in LMS (Figure 2, Table 2) [16]. Nuclear α-thalassemia/mental retardation X-linked (ATRX) loss was reported in around 15% and as high as 30% in ULMS [52]. ATRX is encoded by a gene located on the X chromosome. It belongs to the SWI/SNF (switch/sucrose nonfermentable) family of chromatin remodeling proteins and correlates with defined molecular changes such as the alternative lengthening of telomeres (ALT) phenotype, PDGFRα amplification, and tp53 mutation. Although ATRX is not commonly a driver mutation, it can induce genomic instability leading to further genetic rearrangements and/or mutations. The ALT phenotype is associated with aggressive histologic features, loss of ATRX expression, and poor clinical outcome in LMS [53]. A better understanding of the ALT pathway may help to develop novel therapeutic strategies based on ALT as a target. ULMS also shows BRCA mutation [54].

PTC596 is a first-in-class, oral investigational drug that reduces the levels of BMI1 (B cell-specific Moloney murine leukemia virus integration site 1), required for cancer stem cell survival. BMI-1 is connected to several signaling pathways, including Wnt, Akt, Notch, Hedgehog, and receptor tyrosine kinase pathways. PTC596 acts by binding to tubulin causing a G2/M cell-cycle arrest [55]. A phase I study is investigating PTC596 with dacarbazine in LMS (NCT03761095) [39].

Immunotherapy is being investigated in ongoing trials (Table 3) [39]. A phase II open-label study is evaluating the combination of rucaparib and anti-PD1 antibody nivolumab (NCT04624178). DNA-damaging agents can synergize with immunotherapy by promoting neoantigen release, increasing tumor mutational burden, and enhancing PD-L1 expression [56,57]. Therefore, this combination seems promising in several malignancies.

A phase II study is evaluating the monoclonal anti-PD-L1 antibody avelumab with gemcitabine in a second-line setting (EAGLES NCT03536780).

Several phase II studies are underway in LMS to investigate chemotherapeutic combinations such as eribulin with gemcitabine (phase II, NCT03810976), mixed combinations with a multitarget TKI such as cabozantinib plus temozolomide (phase II, NCT04200443), pazopanib vs. pazopanib plus gemcitabine in ULMS (phase II randomized, PazoDoble trial NCT02203760), or chemotherapy-free combinations such as ribociclib and everolimus (phase II, NCT03114527) [39].

AL3818 (anlotinib) hydrochloride is a multitargeted receptor tyrosine kinase inhibitor with potential FGFR inhibitory activity and antiangiogenesis activity along with a favorable safety profile in a broad range of malignancies. Preliminary data showed promising results, with an interesting progression-free rate at 12 weeks in several STS [58].

This result opened the avenue to the investigation of combination therapy of anlotinib with apatinib. Apatinib is a tyrosine kinase inhibitor that potently and highly selectively inhibits the tyrosine kinase activity of VEGFR2 in vitro and inhibits the activities of VEGFR1, Kit, c-SRC, and RET tyrosine kinases [59]. In a retrospective study, this combination achieved an ORR of 34% with a disease control rate of 69% and a median PFS time of about 8 months [60].

In a phase III trial APROMISS trial (NCT03016819), conducted in LMS, advanced alveolar soft-part sarcoma, and synovial sarcoma, anlotinib was tested as a single agent vs. dacarbazine and placebo in two arms.

Ataxia-telangiectasia and Rad3-related protein (ATR) is an essential DNA damage response regulator and is required for the survival of proliferating cells. ATR repairs damaged DNA. A potent and selective small-molecule ATR inhibitor, which is known as berzosertib (VX-970 or M6620), is under investigation in selected solid tumors including LMS (phase II, NCT03718091) [39]. Interestingly, the study design includes a translational lead-in phase which, differentiates tumors based on mutational status, by next-generation sequencing.

Other studies are currently enrolling various STS subtypes including LMS. A phase I/II study is evaluating the multiple kinase inhibitor lenvatinib with eribulin (LEADER trial, NCT03526679) [39].

Some studies involve immune checkpoint inhibitors plus a chemotherapeutic agent. Given the objective response rate of 8–19% of gemcitabine in LMS, a phase I/II study is investigating its association with pembrolizumab (phase I/II GEMMK NCT03123276). Pembrolizumab is also being investigated in combination with eribulin in a phase II study NCT03899805 and with metronomic cyclophosphamide in a phase II Pembrosarc study (NCT02406781), both including various STS [39].

A basket combination study of inhibitors of DNA damage response, angiogenesis and programmed death ligand 1 (Durvalumab) in patients with advanced solid tumors (phase II, DAPPER trial NCT03851614) is ongoing. In this study, only LMS were enrolled among STS. The primary endpoint was the evaluation of changes in genomic and immune biomarkers.

DCC 3014 is a colony-stimulating factor-1 receptor (CSF1R/c-FMS) inhibitor, which has been evaluated in combination with avelumab in various STS (phase I NCT04242238).

## 4. Rhabdomyosarcoma (RMS)


RMS shows skeletal muscle differentiation and represents less than 3% of adult STS, whilst being the most frequent STS subtype under age 10. Different subtypes can be recognized: alveolar RMS (ARMS), embryonal RMS (ERMS), and pleomorphic RMS (PRMS) [18]. The first two subtypes are more frequent in children, while the latter is common in adults. The outcome is different in pediatric vs. adult patients. The remission rate in children with localized RMS reaches 70%, whereas the prognosis of adults remains poor. There is a known established link between RMS and cancer predisposition syndromes, such as the Li-Fraumeni and neurofibromatosis [61].

Chromosome translocation is typical only of ARMS which in most cases express PAX-FKHR fusion protein (Figure 2, Table 2). This protein represents the transcriptional result of the translocation of a PAX family member (PAX3, PAX7) ordinarily coding for tissue-specific TFs to the FKHR gene, which codes for Forkhead box protein O1 (FOXO1). FOXO1 is a TF that plays an important role in the regulation of gluconeogenesis and glycogenolysis. The PAX3-FOXO1 and PAX7-FOXO1 fusions are products of the characteristic translocations t(2;13)(q35;q14) or t(1;13)(p36;q14), respectively. The fusion protein in ARMS maintains the myogenic lineage but inhibits terminal differentiation [62]. PRMS has a complex karyotype, while ERMS shows chromosomal abnormalities including aneuploidy and polyploidy [16]. Hedgehog signaling, which plays a relevant role in differentiation, has been linked to the development of ERMS [63]. Despite the lack of fusion proteins, ERMS and PRMS overexpress PAX3 and FOXO1. Other alterations found in RMS are p53 mutations, occurring in 0.02–15% of patients, and *MDM2* amplification was described in less than 10% of cases [16]. In ERMS, these percentages are higher [16]. Signaling pathways such as IGF/RAS/MEK/ERK, PI3K/AKT/mTOR, MET, FGFR4, and PDGFR play a role in fusion-positive RMS as well as fusion-negative RMS. Deregulation of the RAS/MEK/ERK/CDK4/6 and G2/M-mitotic spindle checkpoint pathways have also been reported [63].

Although a promising therapeutic strategy against ARMS may be represented by direct inhibition of the mentioned fusion proteins [64], pharmacological agents against these potential targets are lacking. A different approach may be based on influencing fusion protein transcriptional activity through phosphorylation or inhibiting coactivators. Inhibitors of signaling pathways triggered by PAX-FOXO may also be applicable. The main pathways are related to PDGFR, IGFR1, FGFR4, MET, and ALK.

Efforts have been made to develop cancer vaccines that specifically target PAX3-FOXO1 [65]. Otherwise, some of the PAX3-FOXO1 targets, such as FGFR4, CXCR4, and IGF1R, are cell surface antigens and might be blocked by antibodies [65].

A phase II study is exploring erdafitinib, a small-molecule inhibitor of FGFR (NCT03210714) in RMS [39]. Ganitumab is an insulin-like growth factor 1 receptor (IGF-1R) antibody and is under evaluation in a phase 1–2 study (NCT03041701). A phase I trial NCT04299113 is currently investigated mocetinostat with vinorelbine in young patients with RMS [39]. Mocetinostat is a small molecule that inhibits class 1 Histone deacetylases (HDAC), specifically HDAC 1, 2, and 3, resulting in epigenetic changes in tumors.

## 5. Fibroblastic/Myofibroblastic Tumors: Myxofibrosarcoma, Malignant Solitary Fibrous Tumor


Fibroblastic/myofibroblastic tumors present limited specific information with potential clinical drawbacks. Myxofibrosarcoma is characterized by tp53 mutations in more than 40% of the cases [66,67].

A high level of genomic complexity with a recurrent amplification of the chromosome 5p region was also reported (Figure 2, Table 2) [68]. A role for integrin-α10 was found in cell models and may have prognostic relevance in myxofibrosarcoma [69].

Gene amplification on chromosome 5p produces overexpression of TRIO and RICTOR which in turn are also the target of integrin-alpha10. TRIO is a large protein that functions as a GDP to GTP exchange factor and promotes the reorganization of the actin cytoskeleton, thereby playing a role in cell migration and growth. Current ongoing clinical trials investigate immune checkpoint inhibition in this tumor. A phase II study (ENVASARC trial, NCT04480502) with a novel, single-domain PD-L1 antibody that is administered by subcutaneous injection named envafolimab (KN035) alone or combined with ipilimumab, is ongoing [39].

The solitary fibrous tumor has a distinctive translocation which produces a NAB2-STAT6 fusion [70]. This protein consists of the truncated repressor domain of NGFI-A-binding protein 2 (NAB2) and the intact activation domain of STAT6. Recently, cell proliferation through EGR-1 transcriptional expression and IGF2 was found to be induced by the fusion [71]. TERT promoter and p53 mutations have been also found to be associated with malignant transformation of solitary fibrous tumor [70].

## 6. Malignant Tenosynovial Giant Cell Tumor (TGCT)


TGCT is a rare, locally aggressive tumor that arises in the synovium, deserving the name of true synovial sarcoma. The hallmark of TGCT is the aberrant expression of colony stimulating factor (CSF)1 due to the genomic alterations at the CSF1 gene locus on chromosome 1p13 (Table 2) [72]. CSF1 produced by a minority of cells behaves as recruiting factor for other cells which are predominant in this cancer.

Although surgery represents the mainstay of treatment, recurrence is frequent and difficult to be managed with repeated surgical interventions, which highlights the need for active systemic treatment. The landscape is changed with the introduction of pexidartinib (PLX3397). Pexidartinib is a novel, orally administered TKI with strong selective activity against the CSF1 receptor (CSF1R) [73]. This drug shows activity with better results at an increasing duration of treatment. DCC-3014 is an investigational orally administered, potent, and highly selective inhibitor of CSF1R. A multicenter phase 1/2, open-label study of DCC-3014 is ongoing and recruiting patients [39].

## 7. Vascular Tumors: Angiosarcoma and Epithelioid Hemangioendothelioma


Angiosarcoma is a tumor arising from endothelial cells and represents 1–2% of all soft-tissue sarcomas [74]. Secondary angiosarcomas are related to lymphoedema or radiation. Cytogenetically, angiosarcoma is characterized by upregulation of vascular-specific receptor tyrosine kinases. Among angiogenesis genes, MYC plays a role especially in secondary angiosarcoma characterized by the frequent amplification of the region at chromosome 8q24 where MYC is located. MYC gene amplification was related to exposure to ultraviolet and sunlight and was found in more than 80% of radiation-induced angiosarcoma cases. The FLT4 gene, which maps to chromosome on 5q35 and encodes for VEGFR3, is reported in approximately 25% of secondary angiosarcomas.

Murali et al. [75] reported that 50% of angiosarcomas show genetic alterations related to the MAPK pathway, including mutations in RAS, BRAF, MAPK1, and NF1 [16]. Tp53 was altered until 50% of such tumors, while *MDM2* is upregulated in more than 60% of cases [16].

TKI targeting angiogenesis-related pathways such as pazopanib and sorafenib have the potential to be more active in angiosarcoma as compared to other STS, although this assumption was not completely confirmed [76].

Carotuximab (TRC105) is a monoclonal antibody to endoglin, an essential angiogenic target highly expressed on proliferating endothelium and both tumor vessels and tumor cells in angiosarcoma. TRC105 used in combination with pazopanib showed promising activity in a phase 1-2 study [77]. Therefore, a phase III ongoing study started to evaluate this combination [78].

Paclitaxel is considered to serve not only as a chemotherapeutic agent but also as an antiangiogenic drug when used on a weekly schedule. Therefore, it is a standard treatment in vascular tumors.

Paclitaxel activity may be improved by its combined use or by the use of derivatives.

Oraxol consists of paclitaxel and HM30181A, a P-glycoprotein inhibitor, combined to increase the oral bioavailability of paclitaxel [79]. This drug is being investigated in a phase I study (NCT03544567) [39].

A retrospective analysis of patients with locally advanced or metastatic angiosarcoma has recently highlighted the activity of checkpoint inhibitors in this histotype [80].

Several immune checkpoint inhibitors are studied in angiosarcoma. A phase II combination of paclitaxel-avelumab (ASAP trial, NCT03512834) is ongoing.

The combination of immunotherapeutics is particularly appealing in angiosarcoma. Several phase II studies including novel antibodies are ongoing. As an example, the combination of AGEN 2034 (Balstilimab) and AGEN 1884 (Zalifrelimab), which are, respectively, a fully human immunoglobulin (IgG)-4 monoclonal antibody antagonist targeting PD-1 and a human IgG1 anti-cytotoxic T lymphocyte antigen-4 antibody, has been tested in NCT04607200 [39,81]. Another novel combination includes oleclumab and durvalumab (DOSa phase 2 NCT04668300) in various recurrent, refractory, or metastatic sarcoma. Oleclumab is a human monoclonal antibody that binds to CD73 and inhibits the production of immunosuppressive adenosine [82].

Nivolumab plus ipilimumab (phase II NCT02834013) and nivolumab plus sunitinib (phase ½ NCT03277924) are also recruiting active trials [39]. Results of the latter study showed a 6-month PFS of 48% [83].

Epithelioid hemangioendothelioma (EHE) is a low-grade malignant vascular tumor with an intermediate clinical behavior between benign hemangiomas and high-grade angiosarcomas [84]. In most cases, the WWTR1-CAMTA1 fusion protein is identified (Figure 2, Table 2). WWTR1 is also named TAZ and is a transcriptional coactivator and end effector of the Hippo tumor suppressor pathway. CAMTA1 is a putative tumor-suppressive TF [85]. A second fusion event producing a fusion between the YAP and TF E3 (TFE3) genes (YAP-TFE3) occurs in around 10% of all EHEs [86]. Strategies to target the fusion protein represent an exciting promise in EHE as in another STS with known translocation. MEK/MAPK inhibitors can actively reduce both YAP and TAZ levels. This is the rationale behind using trametinib in a phase 2 ongoing study (NCT03148275).

Some studies suggested a role for PI3KCa/Akt/mTOR pathway, especially in hemangioendothelioma.

Rapamycin and its derivatives are promising therapeutic agents with both immunosuppressant and antitumor properties. These rapamycin actions are mediated through the specific inhibition of the mTOR protein kinase. Rapamycin displayed activity in hemangioendothelioma [87].

## 8. Malignant Peripheral Nerve Sheath Tumors (MPNSTs)


MPNSTs are aggressive, frequently metastatic sarcomas that are associated with neurofibromatosis type 1 (NF1), a prominent inherited genetic disease in humans [88].

Half of all MPNSTs develop in individuals with NF1, with a 5-year survival of about 20–50% [89]. NF1 is located on chromosome 17 and codes for neurofibromin, a GTPase-activating protein with negative regulatory activity on RAS/MAPK pathway activity.

Somatic changes in NF1, CDKN2A/B, and PRC2 are found in most MPNSTs, but they are genomically complex (Table 2) [90].

Two multiprotein complexes, the polycomb repressive complex 2 (PRC2) and the switch/sucrose nonfermentable (SWI/SNF) chromatin remodeler act in the control of chromatin status during development and homeostasis [91].

Immunohistochemical loss of SMARCB1 (BAF47)/INI1 expression was found in 70% of cases [92]. The switch/sucrose nonfermentable (SWI/SNF) complex is a highly conserved multi-subunit complex of proteins encoded by numerous genes mapped to different chromosomal regions. The gene is a strong tumor suppressor, and the protein is part of the ATP-dependent SWI/SNF chromatin remodeling complex allowing the transcriptional machinery to access its targets more effectively (Figure 3). SMARCB1 loss activates the Sonic Hedgehog and the wnt/β-Catenin pathway involved in differentiation and proliferation [93]. Epigenetic dysregulation has been extensively correlated with cancer development, progression, and resistance to therapy [94]. Recently, aberrant expression of SMARCB1/INI1 has been found in various tumors such as epithelioid sarcomas, schwannomatosis, synovial sarcomas, and so on [95]. INI1 counteracts the enzymatic function of inherited SWI/SNF-deficiency and has been linked to several benign syndromic tumors including a subset of familial schwannomatosis (linked to SMARCB1) and multiple meningiomas [96].

When INI1 is absent, EZH2 (enhancer of zeste homolog 2) activity is deregulated. EZH2 is the catalytic subunit of PRC2.

Preclinical studies demonstrated that loss of SMARCB1 led to an EZH2 oncogenic dependency [97].

A significant proportion of MPNSTs exhibit recurrent mutations in EED or SUZ12, key components of the PRC2. SUZ12 is located near NF1 at chromosome 17. NF1 and SUZ12 alterations are crucial events in MPNST pathogenesis. Tumors harboring these genetic lesions lose the marker of transcriptional repression, trimethylation of lysine 27 on histone H3, and have dysregulated oncogenic signaling [98].

As SWI/SNF-deficient cells survive through the compensative PRC2 activity, EZH2 inhibitors have the potential to significantly interfere with cell survival. MPNSTs with PRC loss are sensitive to DNA methyltransferase and histone deacetylase (HDAC) inhibitors [99].

Tazemetostat is an orally available, small-molecule selective inhibitor of EZH2 (Figure 3). A phase II multicenter study (NCT02601950) has been performed enrolling INI1-negative tumors or relapsed/refractory synovial sarcoma (Table 3). Despite several attempts to identify novel targeted drugs, a recent study confirms that the “old” multitargeted tyrosine kinase inhibitor pazopanib is still more efficient than new molecules with a clinical benefit rate at 12 weeks of 50.0%, a median PFS of 5.4 months, and a median OS of 10.6 months [100].

New therapies for NF1-related MPNSTs involve Ras/MAPK pathway inhibitors.

Pexidartinib (PLX3397) previously reported in the chapter related to TGCT is investigated together with the mTOR inhibitor sirolimus (phase II NCT02584647, Table 3) [39]. In phase II, SARC 031 sirolimus is combined with MEK inhibitor selumetinib (AZD6244) (NCT03433183, Table 3) [39].

Sarcoma cells showed glutamine dependence. STS subtypes expressing elevated glutaminase (GLS) levels are extremely sensitive to glutamine starvation [101]. The glutaminase inhibitor telaglenastat hydrochloride (CB-839 HCl) is under study in the BeGIN Study (phase II NCT03872427, Table 3), enrolling patients with tumors harboring defined genetic mutations such as NF1 mutation for MPNST [101].

## 9. Tumors of Uncertain Differentiation


### 9.1. NTRK-Rearranged Spindle Cell Neoplasm, Synovial Sarcoma, Epithelioid Sarcoma, Undifferentiated Round-Cell Sarcoma


#### 9.1.1. NTRK-Rearranged Spindle Cell Neoplasm


Neurotrophic tyrosine receptor kinase (*NTRK*)-rearranged spindle cell neoplasm is a recently described soft-tissue tumor entity that occurs predominantly in children and young adults. The three genes named NTRK1 (chromosome 1q23.1), NTRK2 (chromosome 9q21.33), and NTRK3 (chromosome 15q25.3) code for the tropomyosin receptor kinase (TRK) proteins which are typically involved in normal neuronal development. NTRK gene fusion is considered a primary oncogenic driver and can be targeted with TRK inhibitors. In 2020, a panel of experts reached a consensus for NTRK testing, which is recommended in locally advanced/metastatic infantile fibrosarcoma and inflammatory myofibroblastic tumors because of their known high rate of NTRK fusions [102].

#### 9.1.2. Synovial Sarcoma (SS)


SS represents 8–10% of all STS cases. SS is a tumor with a strong driver translocation t(X;18) (p 11.2; q11.2) which produces different SS18:SSX fusion proteins (Figure 2). SS18-SSX participates in chromatin remodeling complexes (Figure 4). During normal transcriptional regulation in mesenchymal cells, the SMARCB1/INI1 or BAF complex coordinates gene expression patterns from both enhancers and promoters, whereas PRC2 activity suppresses the expression of inappropriate gene products (Figure 3). The SS18-SSX fusions competitively replace the wild-type SS18 in the BAF complex and form an altered complex lacking the tumor suppressor BAF47 (hSNF5) (Figure 4). The altered complex binds the Sox2 locus and reverses polycomb-mediated repression, resulting in Sox2 activation [103]. SOX genes encode a family of transcription factors that bind to the minor groove in DNA.

A loss of SMARCB1/INI1 protein is shown in almost all epithelioid sarcoma (ES) and 50% of MPNSTs. A reduced SMARCB1/INI1 protein was shown in 70% of SS in a case series [104].

Aberrant activation of the Wnt/β-catenin pathway is present in most synovial sarcomas [105].

When INI1 is absent, EZH2 activity is deregulated. EZH2 is the catalytic subunit of PRC2. SS has been shown to have high levels of EZH2 expression. Therefore, EZH2 inhibitors such as tazemetostat have been tested in SS. The phase II multicenter-multicohort study (NCT02601950) has been evaluated INI1-negative tumors or relapsed/refractory synovial sarcoma (Table 3). Enrollment is now closed [39].

Anlotinib (AL3818) is an oral small-molecule inhibitor of multiple receptor tyrosine kinases, with a broad spectrum of inhibitory effects on tumor angiogenesis and growth. Anlotinib has shown antitumor activity on STS in preclinical and phase I studies. In a series of STS patients progressing after anthracycline-based chemotherapy, which included SS, promising activity was shown [59]. In a phase III active trial, anlotinib is compared to dacarbazine in advanced alveolar soft-part sarcoma, LMS, and SS (Table 3) (APROMISS trial, NCT03016819) [39].

Most cases of SS have a low tumor mutational burden (TMB), but occasionally a high TMB may be present, explaining the 10% response rate to checkpoint immunotherapy observed in clinical trials in patients with SS [106].

SS is considered a cold tumor. Cancer-testis antigens (CTAs) represent a class of tumor-associated proteins defined based on their tissue-restricted expression to the testis or ovary germline cells and frequent ectopic expression in tumor tissue. CTAs such as NY-ESO-1, PRAME, MAGEA4, and MAGEA1 are expressed at high percentages in SS [107]. Adoptive cell therapies targeting the cancer-testis antigen New York esophageal squamous cell carcinoma-1 (NY-ESO-1) have shown encouraging results [108].

LV305 is a modified, third-generation, nonreplicating, integration-deficient lentivirus-based vector designed to selectively transduce dendritic cells in vivo. LV305 induces expression of the NY-ESO-1 cancer-testis antigen in dendritic cells, promoting immune responses against NY-ESO-1-expressing tumors. A favorable safety profile and clinical activity were evidenced in patients with advanced cancer [109].

#### 9.1.3. Epithelioid Sarcoma


ES is a rare (less than 1% of STS), aggressive soft-tissue neoplasm of uncertain differentiation. It has been characterized by multifocal disease at presentation, local recurrence, and regional metastasis. The hallmark of ES is the loss of SMARCB1/INI1 protein expression which is found in over 80% of cases (Figure 3) [110]. As highlighted in the paragraph of MPNST, the gene is a strong tumor suppressor, and the protein is part of the ATP-dependent SWI/SNF chromatin remodeling complex. Preclinical in vitro and in vivo models show that loss or dysfunction of INI1 can lead to aberrant EZH2 activity or expression. EZH2 contributes to histone methylation inducing a gene transcriptional repressive status of various genes [111,112].

The most relevant results in ES derive from a phase 2 multicenter basket study with tazemetostat which enrolled different INI1-negative solid tumors or synovial sarcoma. Among 62 patients with ES, an ORR of 15% was reported, median PFS reached 5.5 months, and median OS was 19.0 months [113]. These data led to the approval by the FDA of tazemetostat in metastatic or advanced unresectable ES in 2020.

Advancements of these results will derive from a phase Ib/III trial of tazemetostat in combination with doxorubicin as frontline therapy (NCT04204941) (Table 3) [39]. A case report of response at the progression of tazemetostat was reported with ipilimumab and nivolumab [114]. A phase II study with this combination (NCT04416568) is currently recruiting INI1-negative tumors and among them ES (Table 3) [39].

#### 9.1.4. Undifferentiated Small Round-Cell Sarcoma


According to the 2020 WHO Classification, undifferentiated small round-cell sarcomas include: Ewing sarcoma, round-cell sarcomas with EWSR1-non-ETS fusions, CIC-rearranged sarcomas, and BCOR-rearranged sarcomas.

Ewing sarcoma is the second most common pediatric bone tumor. Patients with metastatic Ewing sarcoma have a 5-year survival between 18% and 30%. A specific translocation involving EWS on chromosome 22 with one of the E26 transformation-specific transcription factory family genes characterized this cancer. The EWS-FLI1 fusion gene, t(11;22)(q24;q12) is found in ∼85% of Ewing sarcoma [115]. Fusion protein act as a TF and potent oncogene. EWS-FLI1 increases the expression of many downstream targets involved in tumor survival and growth, for example, IGF1, GLI1, Myc, and ID2, and decreases expression of cell-cycle regulators and proapoptotic genes [115].

Some approaches targeting downstream signaling are in preclinical phases and involve IGF and several RTKs. CD99 antigen is frequently expressed in Ewing sarcoma; therefore, it represents a diagnostic and appealing therapeutic target. These premises label EWS-FLI1 as a key potential target. Recently, TK216, a first-in-class small molecule, has been proposed for Ewing sarcoma. This molecule inhibits the binding between fusion protein and RNA helicase A, thus, inducing cell apoptosis. It is currently investigated in a phase I study (NCT02657005) [39].

Undifferentiated round-cell sarcomas lacking these rearrangements, known as ‘Ewing-like’ sarcomas, usually show atypical clinical presentation and focal CD99 positivity. This group of tumors can be subdivided into capicua transcriptional repressor (CIC)-rearranged sarcomas, Bcl6 corepressor (BCOR)-rearranged sarcomas, sarcomas with EWSR1 fusion to non-ETS family members, and unclassified round-cell sarcomas [116].

CIC-DUX4 fusion sarcoma shows a more aggressive clinical course than the classical Ewing sarcoma. CIC rearranged sarcomas arise in old age and affect preferentially soft tissues [117].

CIC is a tissue-specific transcriptional repressor. It is highly conserved during evolution. CIC regulates several physiological and developmental processes. CIC-DUX4 gene fusion results from either a t(4;19) or t(10;19) translocation [118]. The CIC-DUX4 fusion retains most of the wild-type CIC and might act as an aberrant TF [119].

## 10. Immunotherapy


In the previous paragraphs, a list of clinical studies with immunotherapy was reported for each STS subtype. Here, a comprehensive view of the role of immunotherapy is outlined focusing on potentially more responsive histotypes and promising clinical trials.

Immune responsive tumors are generally characterized by immune infiltration, strong PD-L1 expression, and high TMB. Nearly all STS subtypes showed PD-L1 expression, albeit with a broad range of positivity [120]. Other studies reported PD-L1 expression by immunohistochemistry (≥1%) in some histotypes and not in others. High PDL-1 expression (30%) was shown in undifferentiated pleomorphic sarcomas, lower expression (10%) in DDLPS, and absent expression in WDLPS, MLPS, PLPS, synovial sarcomas, and Ewing sarcomas [121].

High PD-L1 expression was associated with poorer OS in two meta-analyses including bone and STS [122,123].

A general rule is that increased genomic aberrations correlate with increased expression of tumor antigens, which may be potential targets for immunotherapy. Tumor types with higher mutational burden have higher response rates with PD-1/PD-L1 blockade. TMB showed a good correlation to response rate [124].

In a wide study including clinical and genomic data of 1662 advanced cancer patients, higher somatic TMB (highest 20% in each histology) was associated with better OS. For most cancer histology, an association between higher TMB and improved survival was observed. The TMB cut points associated with improved survival varied markedly between cancer types. The authors concluded that there is not a universal definition of high TMB [125].

Immunotherapy may have low chances to succeed in unselected STS, although a subset may benefit from immunotherapy. In fact, hypermutation was detected in tumor types not previously associated with high mutation burden [126].

Furthermore, mismatch repair-deficient sarcomas represent a low rate of all STS (1%) but showed a significantly elevated tumor mutation burden relative to mismatch repair-proficient sarcomas (median 16 vs. 4.6, *p* < 0.001) [127].

In a study on a series of undifferentiated sarcomas, 15% had a high mutational burden that correlated with an immune signature and good prognosis [128].

Attempts to define potentially responsive patients have generated models separating high- from low-risk patients. Low-risk patients showed an immune-infiltrating profile with increased immune cell infiltration (e.g., CD8 T cell and activated natural killer cells), higher expression of immune-stimulating molecules, higher stimulating cytokines and corresponding receptors, higher innate immunity molecules, and stronger antigen-presenting capacity. These features translate into better overall survival time [129]. DNA methylation analysis indicated that relative high methylation was associated with better OS. A recent study based on the composition of the tumor microenvironment identifies an immune-high subtype enriched with B-cells that is particularly responsive to anti-PD1 and showed increased survival [130].

The anti-PD1 pembrolizumab was studied in a multicenter, single-arm, open-label, phase 2 trial which enrolled 86 patients with an ORR of 18%. The activity was documented in undifferentiated pleomorphic sarcoma or DDLPS [131]. An immune signature was found to correlate with better outcomes [132]. Confirmatory results have come from the phase II, nonrandomized, multicenter AcSé Pembrolizumab study which enrolled 80 patients. The ORR was around 16% (partial response) without correlation to histotypes [133].

The activity of anti-PD1 either pembrolizumab or nivolumab and of the combination of anti-PD1 plus anti-CTLA4 (nivolumab plus ipilimumab) was reported [134], with minor enthusiasm for nivolumab alone and better results for the combination [135].

Various studies investigated the combination of anti-PD1 with other agents, such as axitinib in another phase 2 study [136], and showed activity, especially in alveolar soft-part sarcoma. These data relative to a distinct subtype were further confirmed recently [137].

The actual role for immunotherapy in STS is probably better defined for some subtypes such as alveolar soft-part sarcoma and angiosarcoma which need further trials regardless.

Checkpoint inhibitors can activate the endogenous response in some immune responsive STS, but cold tumors are refractory to these therapeutic strategies because of the absence/reduced expression of neoantigens, defective antigen presentation, and microenvironment interferences. For these tumors, cellular immunotherapy represents a powerful tool to be investigated.

Chimeric antigen receptors (CARs) are receptors that have been engineered to give T cells the new ability to target a specific protein. CAR T cells are T cells that have been genetically engineered with CARs to target and destroy tumor cells expressing a particular antigen more effectively. Several sarcoma-associated antigens amenable to CAR-T cell treatment have recently emerged with encouraging results. These include cancer-testis antigens, HER2, GD2, IL-11RA, FAP, B7-H3, CD44v6, IGF-1R, and ROR1 [138]. As in hematological malignancies, a favorable cytokine profile induced by lymphodepletion appears to be necessary for CAR-T therapy probably because lymphodepleting conditioning eliminates regulatory T cells [139]. ADP-A2M4 (MAGE-A4) SPEAR T-cell therapy was investigated in a phase II study (NCT04044768) in synovial and myxoid liposarcoma patients who had received prior chemotherapy and whose tumors express the MAGE-A4 tumor antigen [39]. Similarly, NY-ESO-1 T cells are in a phase II study in advanced myxoid/round-cell LPS (NCT02992743) [39].

## 11. Conclusions


STS represents a challenge and an opportunity for geneticists and medical oncologists. Precision medicine has already come into the complex and heterogeneous field of sarcoma. In a study of 584 patients with STS in the American Association for Cancer Research (AACR) Project Genomics Evidence Neoplasia Information Exchange (GENIE) database, a genetic alteration with the potential to influence therapy was identified in 41% of cases [140]. In a large database of 5635 adult and pediatric, bone, and STS, 107 patients with matching clinical data were found. Among these, 57% had actionable mutations and 30% were enrolled in clinical trials [141].

In this scenario, there are ongoing observational studies to collect potentially relevant pathological and clinical data. Among these, the Sarcoma Biology and Outcome Project (SarcBOP- NCT04758325) is a prospective registry study on biological disease profile, intervention type, and clinical outcome with an estimated enrollment of 3000 patients by December 31, 2032, the planned study completion date [39]. The length of this study gives a prospect of the efforts required for STS.

Clinical studies are increasingly enriched with the evaluation of genomic and immune changes performed at baseline and during treatment to dynamically assess the tumor under selective pressure induced by treatment.

Biological advancements are showing the way to overcome the present undefined middle-ground and shape-tailored strategies in the near future.

## Figures and Tables

**Figure 1 cancers-13-02359-f001:**
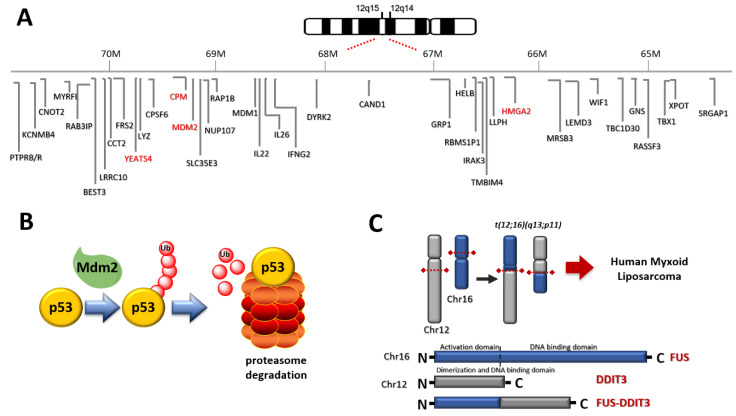
(**A**) Chromosome 12 represents the hallmark of WDLPS and DDLPS. Amplified relevant genes are depicted in red. (**B**) Relationship between Mdm2 and p53: Mdm2 induces degradation of p53 through polyubiquitylation; (**C**) the translocation t(12;16)(q13;p11) induces the expression of FUS DDIT 3 fusion protein in MLPS.

**Figure 2 cancers-13-02359-f002:**
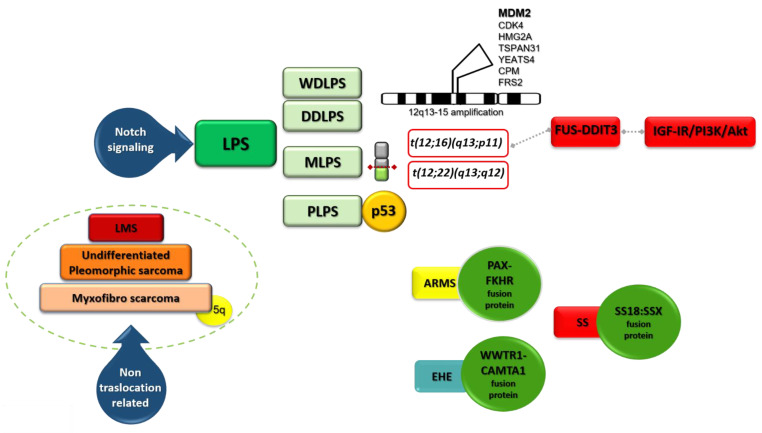
A schematic view of genetic alterations in LPS.

**Figure 3 cancers-13-02359-f003:**
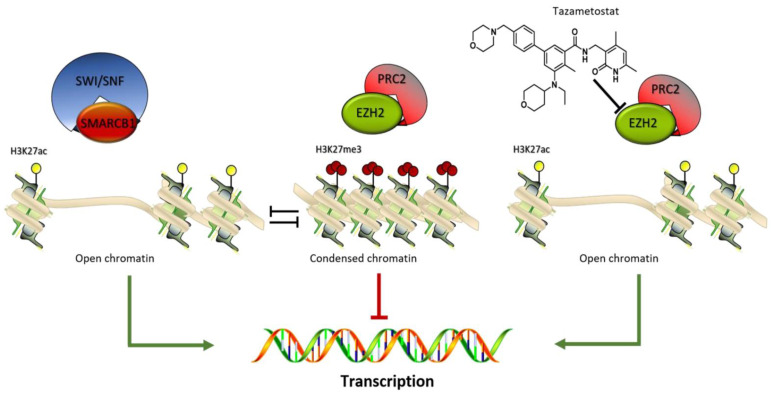
SMARCB1 /PRC2 balance. The SWI/SNF-SMARCB1 binding induces the activation of the epigenetic enzyme such as the histone acetyltransferase inducing a chromatin open-frame mediated by the acetylation of lysine 27 on histone 3 (H3K27Ac) and consequential target genes expression. The PRC2 complex acts as antagonist inducing a gene silencing by an increase in the methylation status of H3K27me3. This defined equilibrium is modified by the selective EZH2 inhibitor tazemetostat which inhibiting the enzyme activity, blocks the PRC2 complex silencing effect.

**Figure 4 cancers-13-02359-f004:**
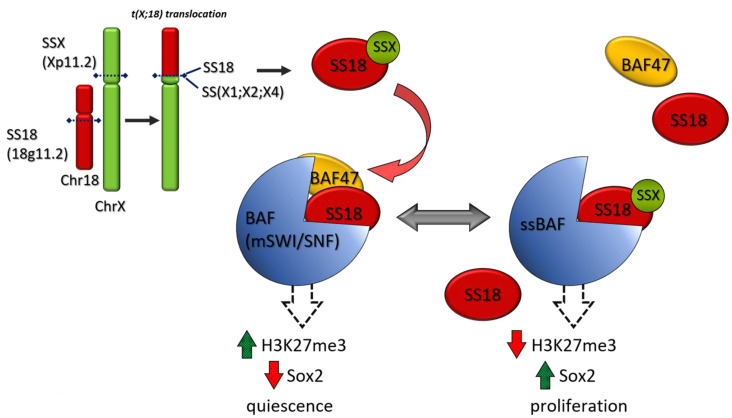
SS 18 SSX fusion oncoproteins in SS. The common translocation in SS, t(X;18)(p 11.2; q11.2), leads to the expression of SS 18 SSX fusion proteins. In normal mesenchymal cells, BAF(mSWI /complex induces a cell quiescence by an increase in “repressive” epimarks (H3K27me3) and Sox 2 inhibition. In SS, the SS 18 SSX fusion proteins replace the wild type SS 18 in the BAF complex and form an altered complex lacking the tumor suppressor BAF 47 (hSNF 5). The altered complex binds Sox 2 causing its activation and cell proliferation.

**Table 1 cancers-13-02359-t001:** Most common malignant soft-tissue tumors.

Malignant adipocytic tumors (15–25%)	Well-differentiated LPS Dedifferentiated LPS Myxoid LPS Pleomorphic LPS Myxoid pleomorphic LPS
LMS (5–10%)	
RMS (<3%)	Alveolar RMS Embryonal RMS Pleomorphic RMS
Fibroblastic/myofibroblastic tumors	Myxofibrosarcoma Malignant solitary fibrous tumor
TGCT	
Vascular tumors	Angiosarcoma (1–2%) Epithelioid hemangioendothelioma
MPNST	
Tumors of uncertain differentiation	NTRK-rearranged spindle cell neoplasm SS (8–10%) ES (<1%)
Undifferentiated small round-cell sarcoma	Ewing sarcoma Round-cell sarcoma with EWSR1-non-ETS fusions CIC-rearranged sarcomas BCOR-rearranged sarcomas

Legend: LPS: liposarcoma, LMS: leiomyosarcoma, RMS: rhabdomyosarcoma, TGCT: tenosynovial giant cell tumor, MPNST: malignant peripheral nerve sheath tumors, SS: synovial sarcoma, and ES: epithelioid sarcoma.

**Table 2 cancers-13-02359-t002:** Translocations and most common mutations in STS.

Tumor Type	Translocations/Fusion Protein (Reported Incidence)	Genetic Aberration	Receptor Overexpression	Pathways
Malignant adipocytic tumors		p53 (10–20%)		Notch signaling
- Well-differentiated LPS		amplification *12q13-15* amplification of MDM2, CDK4, HMG2A, TSPAN31, YEATS4, CPM	MET, IGFR, AXL, EGFR	
- Dedifferentiated LPS		amplification *12q13-15* amplification of MDM2, CDK4, HMG2A, TSPAN31, YEATS4, CPM aurora kinase	MET, IGFR, AXL, EGFR	
- Myxoid LPS	t(12;16)(q13;p11) FUS-DDIT3 (90%) t(12;22)(q13;q12) EWSR1-DDIT3 (5%)	PI3K/Akt (26%)		
- Pleomorphic LPS		p53 (60%)		
LMS ULMS		P53 Rb1 ATRX loss BRCA		
RMS		P53 MDM2		IGF/RAS/MEK/ERK PI3K/AKT/mTOR MET, FGFR, PDGFR
- ARMS	T(2;13)(q35;q14), t(1;13)(p36;q14) PAX-FKHR fusion protein			
- ERMS				Hedgehog signaling
- PRMS				
Fibroblastic/myofibroblastic tumors				
- Myxofibrosarcoma		P53 (>40%) Amplification of 5q region		
- Solitary fibrous tumor	inv12(q13q13) NAB2-STAT6 fusion protein	P53 TERT promoter		
Malignant tenosynovial giant cell tumor	1p13 (CSF)			
Vascular tumor				
- Angiosarcoma		P53 (50%) MDM2 (>60%) Amplification 8q24 Amplification 5q35		MYC FLT MAPK (RAS, BRAF, MAPK1, NF1)
- EHE	t(1;3)(p36.3;q25) WWTR1-CAMPTA1 fusion protein YAP-TFE3			PI3KCa/Akt/mTOR
MPNST		Loss of SMARCB1/INI1 Mutations EED/SUZ12		Sonic Hedgehog pathway Wnt/β-catenin pathway
Tumors of uncertain differentiation				
- SS	t(X;18) (p 11.2; q11.2) → SS18:SSX fusion proteins			Wnt/β-catenin pathway
- ES		Loss of SMARCB1/INI1		
Undifferentiated small round-cell sarcoma				
- Ewing sarcoma	t(11;22)(q24;q12) → EWS-FLI1 fusion protein			
- Round-cell sarcomas with EWSR-non-ETS fusions				
- CIC-rearranged sarcomas	t(4;19) or t(10;19) translocation → CIC-DUX4 fusion protein			
- BCOR-rearranged sarcomas				

Legend: LPS: liposarcoma, WDLPS: well-differentiated LPS, DDLPS: dedifferentiated LPS, MLPS: myxoid LPS, PLPS: pleomorphic LPS, LMS: leiomyosarcoma, ULMS: uterine leiomyosarcoma, RMS: rhabdomyosarcoma, ARMS: alveolar RMS, ERMS: embryonal RMS, PRMS: pleomorphic RMS, TGCT: tenosynovial giant cell tumor, MPNST: malignant peripheral nerve sheath tumors, EHE: epithelioid hemangioendothelioma, SS: synovial sarcoma, and ES: epithelioid sarcoma.

**Table 3 cancers-13-02359-t003:** Summary of most relevant clinical studies grouped by histotypes [39].

Trial Identifier	STS Histotype	Drugs	Phase	Study Hallmarks	Treatment Arms	Estimated Enrollment	Status
NCT01636479	LPS	SAR405838	I				Completed
NCT01463696		MK 8242	I				Completed
NCT01209598		Palbociclib	II				Completed
NCT03114527	LPS-LMS	Ribociclib everolimus	II				Active
NCT04438824	LPS	Palbociclib Anti-PD-1	II				
NCT02606461 (SEAL)	LPS	Selixenor	II/III		Selixenor placebo	342	Active not recruiting
NCT02978859	LPS	Sitravanib	II				Active
NCT03761095	LMS	PTC596 Dacarbazine	I				Active
NCT04242238	STS	Avelumab DCC-3014	I				Active
NCT03526679 (LEADER)	LMS LPS Adult STS Advanced cancer	Lenvatinib Eribulin	I/II				Active
NCT03123276 (GEMMK)	LMS UPS	Pembrolizumab Gemcitabine	I/II				
NCT04624178	LMS	Rucaparib Nivolumab	II				Active
NCT03536780 (EAGLES)	LMS	Avelumab Gemcitabine	II				Active
NCT03810976	LMS	Eribulin Gemcitabine	II				Active
NCT04200443	LMS	Cabozantinib Temozolamide	II				Active
NCT02203760 (PazoDoble)	ULMS	Pazopanib Gemcitabine	II randomized		Pazopanib Pazopanib+Gemcitabine	107	Active
NCT03114527	DDLPS LMS	Ribociclib Everolimus	II	Pretreated			Active
NCT03851614 (DAPPER)	LMS MMRp-CRC PA	Durvalumab Olaparib Cediranib	II randomized	Basket study	Durvalumab+Olaparib Durvalumab+Cediranib	90	Active
NCT03718091	LMS Osteosarcoma Solid tumors	MSS20	II			223	Active not recruiting
NCT03899805	LMS LPS UPS	Eribulin Pembrolizumab	II				Active
NCT02406781 (PEMBROSARC)	LMS STS	Metronomic CP Pembrolizumab	II				Active
NCT03016819 (APROMISS)	LMS SS ASPS	AL3818 Dacarbazine (DTIC)	III	Pretreated LMS/SS	ASPS AL3818 LMS/SS AL3818 vs. DTIC LMS AL3818 vs. placebo	325	Active
NCT04480502 (ENVASARC)	UPS Myxofibrosarcoma	Envafolimab Ipilimumab	II randomized		Envafolimab Envafolimab+Ipilimumab	160	Active
NCT03512834 (ASAP)	Angiosarcoma	Paclitaxel Avelumab	II				Active
NCT04607200	Angiosarcoma	AGEN2034 AGEN1884	II				Active
NCT03277924 (ImmunoSarc)	STS Bone tumors		I/II			270	Active
NCT02834013	Rare tumors		II				
NCT02601950	INI-1 negative tumors SS	Tazemetostat	II			250	Active
NCT02584647	Sarcoma MPNST	PLX3397 Sirolimus	I/II				Active
NCT03433183 (SARC031)	MPNST	Selumetinib (AZD6244) Sirolimus	II				Active
NCT03872427 (BeGIN)	NF1 Aberrations, NF1 Mutant MPNST, KEAP1/NRF2, and LKB1 Aberrant Tumors	Telaglenastat Hydrochloride	II	basket		108	Active
NCT04204941	STS ES	Tazemetostat Doxorubicin	Ib/III		Tazemetostat + Doxorubicin Doxorubicin + Placebo	164	Active
NCT04416568	INI1-negative tumors	Nivolumab Ipilimumab	II				Active

Legend: LPS: liposarcoma, LMS: leiomyosarcoma, DDLPS: dedifferentiated LPS, MMRp-CRC: mismatch repair proficient colorectal cancer, PA: pancreatic cancer, UPS: undifferentiated pleomorphic sarcoma, ULMS: uterine leiomyosarcoma, RMS: rhabdomyosarcoma, ARMS: alveolar RMS, ERMS: embryonal RMS, PRMS: pleomorphic RMS, TGCT: tenosynovial giant cell tumor, MPNST: malignant peripheral nerve sheath tumors, EHE: epithelioid hemangioendothelioma, SS: synovial sarcoma, ES: epithelioid sarcoma, and ASPS: alveolar soft-part sarcoma.

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
