# Peer review of "Toward a Personalized Therapy in Soft-Tissue Sarcomas: State of the Art and Future Directions"

_cancers, 2021, doi:10.3390/cancers13102359_

Round 1
Reviewer 1 Report
Congratulations to the authors. A very compreshensive and updated review of the current treatment and clincial trials for advanced STS.
Author Response
Thank you very much for the favorable opinion expressed.
Reviewer 2 Report
- Well-written and representative review of the current therapeutic options regarding soft tisse sarcomas (STS)
- The structure of the manuscript is good providing the most common mutations and their pathogenetic mechanism for each subtype of STS
- The chapter of immunotherapy could be developed more but still provides a nice comprehensive view of the role for immunotherapy.
- Well-represented tables of the common mutations of STS, and the most relevant clinical trial as well.
- Well-illustrated pictures.
- The references are complete and updated.
Author Response
Thank you very much for the favorable opinion expressed.
- The chapter of immunotherapy could be developed more but still provides a nice comprehensive view of the role for immunotherapy.
I agree with you, but an extensive discussion concerning immunotherapy in STS was beyond the aim of our study.
Reviewer 3 Report
The authors well showed the overviews of precision medicine to sarcoma treatment; there are some points to be added for better manuscript, I think.
・L54-61: In this paragraph, clinical trial data of trabectedin were shown and referred to the approval in FDA; however, trabectedin had been approved in Europe in 2007 and the practical data showed some important information such as the efficacy to myxoid liposarcoma. These data should be added in the paragraph or the section of liposarcoma.
・L281-289: Clinical trails of selinexor to liposarcoma were shown in the paragraph; please add the reference of the phase 2-3 trial (NCT02606461), which was presented in the CTOS 2020.
・L627-45: In the section of epithelioid sarcoma, ENH2 targeted therapy was described; in the US, FDA approved tazemetostat, EZH2 inhibitor to epithelioid sarcoma in 2020, and it should be also described in the section.
Author Response
Thank you very much for the favorable opinion expressed.
- As concerns your 1st suggestion “L54-61: In this paragraph, clinical trial data of trabectedin were shown and referred to the approval in FDA; however, trabectedin had been approved in Europe in 2007 and the practical data showed some important information such as the efficacy to myxoid liposarcoma. These data should be added in the paragraph or the section of liposarcoma.”:
At L66-70, I introduce the following paragraph: “Initial information coming from a compassionate-use program of trabectedin documented activity in advanced pretreated myxoid LPS (MLPS) [7]. Since trabectedin approval in Europe in 2007, an expanding amount of data have been supporting its efficacy in real-world evaluations [8]. Recent data confirm the efficacy of trabectidin in patients with LPS and LMS with higher PFS in MLPS [9].”
References 7-9 were added:
- Grosso F.; Jones R.L.; Demetri G.D.; Judson I.R.; Blay J.Y.; Le Cesne A.; Sanfilippo R.; Casieri P.; Collini P.; Dileo P.; et al. Efficacy of trabectedin (ecteinascidin-743) in advanced pretreated myxoid liposarcomas: a retrospective study. Lancet Oncol. 2007;8:595-602. doi: 10.1016/S1470-2045(07)70175-4.
- de Sande González L.M.; Martin-Broto J.; Kasper B.; Blay J.Y.; Le Cesne A. Real-world evidence of the efficacy and tolerability of trabectedin in patients with advanced soft-tissue sarcoma. Expert Rev Anticancer Ther. 2020;20:957-963. doi: 10.1080/14737140.2020.1822744.
- Kobayashi H.; Iwata S.; Wakamatsu T.; Hayakawa K.; Yonemoto T.; Wasa J.; Oka H.; Ueda T.; Tanaka S. Efficacy and safety of trabectedin for patients with unresectable and relapsed soft-tissue sarcoma in Japan: A Japanese Musculo-skeletal Oncology Group study. Cancer 2020;126:1253-1263. doi: 10.1002/cncr.32661.
- L281-289: Clinical trials of selinexor to liposarcoma were shown in the paragraph; please add the reference of the phase 2-3 trial (NCT02606461), which was presented in the CTOS 2020.
Reference 51 was added according to your suggestion.
Gounder M.; Razak A.A.; Somaiah N.; MartinBroto J.; Schuetze S.; Grignani G.; Chawla S.P.; Chmielowski B.; Vin-cenzi B.; Silvia Stacchiotti S.; et al. A phase 2/3, randomized, double blind, cross-over, study of selinexor versus placebo in advanced unresectable dedifferentiated liposarcoma (DDLS). Oral presentation at CTOS 2020 Memorial Sloan Kettering Cancer Center, New York (USA) November 20, 2020.
- L627-45: In the section of epithelioid sarcoma, ENH2 targeted therapy was described; in the US, FDA approved tazemetostat, EZH2 inhibitor to epithelioid sarcoma in 2020, and it should be also described in the section.
At L679-680, the following phrase was added:
“These data led to the approval by FDA of tazemetostat in metastatic or advanced unresectable ES in 2020.”
Reviewer 4 Report
This is a very extensively researched manuscript bringing together a broad available knowledge on mutations in sarcoma. It focusses on potential druggable targets. In the introduction also local standard treatments like surgery and radiotherapy are mentioned. Here I have a comment concerning Page 2, line 44-47: radiotherapy is like surgery a cornerstone of soft tissue sarcomas, nearly all sarcomas are treated by radiotherapy besides very small ones not infiltrating the fascicle. Often nowadays radiotherapy is used in the neoadjuvant setting. Radiotherapy can also be curative in the primary setting for example for elderly patients. I would propose that authors extend this paragraph a bit and then consequently delete radiotherapy and surgery treatment remarks later one, like on Page 3, lines 115-117., this would be consequent as later on this is also no longer mentioned. Additionally, this comment is not fully correct.
Author Response
Following the comment “In the introduction also local standard treatments like surgery and radiotherapy are mentioned. Here I have a comment concerning Page 2, line 44-47: radiotherapy is like surgery a cornerstone of soft tissue sarcomas, nearly all sarcomas are treated by radiotherapy besides very small ones not infiltrating the fascicle. Often nowadays radiotherapy is used in the neoadjuvant setting. Radiotherapy can also be curative in the primary setting for example for elderly patients. I would propose that authors extend this paragraph a bit and then consequently delete radiotherapy and surgery treatment remarks later one, like on Page 3, lines 115-117., this would be consequent as later on this is also no longer mentioned. Additionally, this comment is not fully correct.”
At L47-52, the following paragraph “Radiotherapy plays a definite role in several settings of STS. In adjuvant phases may reduce the recurrence risk especially when there are close or infiltrated margins. In the neo-adjuvant setting, the combined use of radio- and chemotherapy produced better results in terms of overall survival in high-risk STS of the extremities [2]. Stereotactic body radiotherapy (SBRT) was shown to compare well to surgery in case of lung metastases [3]. In advanced/palliative setting, radiotherapy may represent a compelling choice” was added.
References 2,3 were added.
- Chowdhary M.; Chowdhary A.; Sen N.; Zaorsky N.G.; Patel K.R.; Wang D. Does the addition of chemotherapy to neoadjuvant radiotherapy impact survival in high-risk extremity/trunk soft-tissue sarcoma? Cancer 2019 ; 125:3801-3809. doi: 10.1002/cncr.32386
- Tetta C.; Londero F.; Micali L.R.; Parise G.; Algargoush A.T.; Algargoosh M.; Albisinni U.; Maessen J.G.; Gelsomino S.; Stereotactic Body Radiotherapy versus Metastasectomy in patients with pulmonary metastases from soft tissue sarcoma. Clin Oncol (R Coll Radiol). 2020 ;32:303-315. doi: 10.1016/j.clon.2020.01.005
At L126, the mention of surgery and radiotherapy was deleted.
Thank you very much for the favorable opinion expressed.
Reviewer 5 Report
The manuscript describes the important topic about the current state and prospective avenues for therapy of soft tissue sarcomas (STS).
I have a couple suggestions about this manuscript:
1) The legend to Figure 1 is not clear and have to be re-written.
2) It will be better if the data illustrating the in vitro effects of the targeted-based drugs will be presented separately from the clinical trials data. For example, the data shown in the lines 278-279 presents mostly the effect of pan-FGFR-inhibitor in vitro, and the effect of erdafinitib was shown for ONE patient. Therefore, to avoid misunderstanding of the audience, it will be better to segregate the in vitro studies from clinical findings.
3) The examples of the perspective therapeutic agents are missing for some of therapies. For example, "Recently, a first-in-class small molecule has been proposed for Ewing sarcoma" (lines 661-662). Is this about the CD99-targeted nanoparticle (CD99-TNP/Ir)?
In general, the manuscript is very well written and covers all the fields, including the genetic profile of various types of STS and modern information about the clinical trials that were recently finished and ongoing, as well. The manuscript will be interesing to a broad audience and can be accepted for publication after minor revision.
Author Response
- The legend to Figure 1 is not clear and has to be re-written. The legend was re-written accordingly.
- It will be better if the data illustrating the in vitro effects of the targeted-based drugs will be presented separately from the clinical trials data. For example, the data shown in the lines 278-279 presents mostly the effect of pan-FGFR-inhibitor in vitro, and the effect of erdafinitib was shown for ONE patient. Therefore, to avoid misunderstanding of the audience, it will be better to segregate the in vitro studies from clinical findings.
Thank you for your suggestion which significantly contributes to a better organization of the text.
Accordingly, the mentioned in vitro studies were separated:
-at L293-295 the results concerning erdafitinib in LPS were anticipated relative to clinical trials.
-at L 409-414 results with PTC596 were anticipated.
- The examples of the perspective therapeutic agents are missing for some of therapies. For example, “Recently, a first-in-class small molecule has been proposed for Ewing sarcoma" (lines 661-662). Is this about the CD99-targeted nanoparticle (CD99-TNP/Ir)?
No, it is not. The small molecule is TK216. Due to the page layout, it was unclear.
Thank you very much for the favorable opinion expressed.